# Sex Differences of the Functional Brain Activity in Treatment-Resistant Depression: A Resting-State Functional Magnetic Resonance Study

**DOI:** 10.3390/brainsci12121604

**Published:** 2022-11-23

**Authors:** Jifei Sun, Yi Luo, Yue Ma, Chunlei Guo, Zhongming Du, Shanshan Gao, Limei Chen, Zhi Wang, Xiaojiao Li, Ke Xu, Yang Hong, Xue Yu, Xue Xiao, Jiliang Fang

**Affiliations:** 1Guang’anmen Hospital, China Academy of Chinese Medical Sciences, Beijing 100053, China; 2Dongzhimen Hospital, Beijing University of Chinese Medicine, Beijing 100700, China; 3Beijing First Hospital of Integrated Chinese and Western Medicine, Beijing 100026, China

**Keywords:** treatment-resistant depression, rs-fMRI, amplitude of low frequency fluctuations, sex, major depressive disorder

## Abstract

The presence of different clinical symptoms in patients with treatment-resistant depression (TRD) of different sexes may be related to different neuropathological mechanisms. A total of 16 male patients with TRD, 18 female patients with TRD, 18 male healthy controls (HCs) and 19 female HCs completed this study. We used the amplitude of low frequency fluctuations (ALFF) method to analyze the results. Moreover, the correlation between abnormal brain areas and clinical symptoms in different sexes of the TRD groups was also analyzed. The effects of the sex-by-group interaction difference in ALFF among the four groups was located in the left middle frontal gyrus, left precentral gyrus and left precuneus. Post hoc comparisons revealed that the male TRD group had lower ALFF in the left middle frontal gyrus and left precentral gyrus compared with the female TRD group. There was a positive correlation between the left middle frontal gyrus, the left precuneus and the 17-item Hamilton Rating Scale for Depression scale (HAMD-17) scores, and a negative correlation between the left precentral gyrus and the HAMD-17 scores in the female TRD group. This study will provide some clinical reference value for the sex differences in neuropathological mechanisms of TRD.

## 1. Introduction

Major depressive disorder (MDD) is the most common psychiatric disorder, and even though MDD receives standardized treatment, 30% of MDD patients are still clinically ineffective or have poor outcomes [1]. We defined this MDD without significant improvement after a full dose and course of antidepressant medication of at least two different mechanisms clinically as treatment-resistant depression (TRD) [2]. Compared to patients with non TRD, the pathogenesis of TRD is complex, and patients with TRD have higher outpatient and inpatient visit rates and 50% higher direct and indirect medical costs, and TRD has more severe anxiety, depression and cognitive impairment, which causes significant distress in patients’ lives [3,4,5].

In addition, TRD is one of the distinct subtypes of MDD, and epidemiological surveys have shown that female are almost two to three times more likely to suffer from depression than male [6,7]. In the higher age groups (>65 years), although the incidence usually decreases in both male and female, it remains significantly higher in female than in male [8]. In addition, most female with TRD have more severe anxiety, depression, and insomnia compared to female with TRD [6,9,10]. This predisposes female with TRD to higher numbers of sedative and hypnotic use than male [11,12]. Therefore, understanding the underlying neuropathological mechanisms of TRD in different sexes is a very important reference for understanding the differences in their clinical symptoms.

In recent years, magnetic resonance imaging (MRI) have been widely used in the field of psychiatric disorders, including TRD, recurrent depressive episode and other subtypes of MDD [13,14,15]. Moreover, MRI studies have been conducted in MDD patients of different sexes to provide some implications for the understanding of the differences in neuropathological mechanisms between the sexes. [16,17,18]. In terms of brain structure, A study found that the surface area (SA) of the left ventrolateral prefrontal cortex was significantly reduced in the female MDD, whereas the SA was increased in the male MDD [18]. Another study found that the gray matter (GM) of the limbic system region had lower in the female MDD, whereas the GM of the striatal region had lower in the male MDD [19]. It was also found that the volumes of the left inferior anterior cingulate were smaller in the male MDD than the female MDD, while the volumes of the amygdala were larger in the female MDD [20].

Resting-state functional MRI (rs-fMRI) is currently an important tool for the study of psychiatric disorders and is able to respond to blood oxygen level dependence (BOLD) [21]. The amplitude of low frequency fluctuations (ALFF) reflects the level of spontaneous activity of the brain on a voxel-by-voxel basis and is a commonly used study method for rs-fMRI, and this method is widely used in the study of MDD, autism, schizophrenia and other psychiatric disorders [22,23,24]. Moreover, some studies have also used ALFF to observe sexes differences of the functional brain activity in MDD patients [14,15]. A study found that ALFF differences between males and females with MDD in some brain regions of the frontoparietal network (middle frontal gyrus, inferior parietal lobule and right precuneus), attention network (left superior temporal pole, left inferior parietal lobule), cerebellum network (left crus 1 of the cerebellum), and auditory network (left postcentral gyrus) [16]. Compared to the female MDD, another study found that ALFF of the posterior cingulate gyrus and bilateral caudate nucleus had lower in the male MDD, suggesting that there may be sexes different of the functional brain activity in MDD [17]. However, to date, studies on the resting state of TRD in different sexes have not been elucidated.

In the present study, we investigated sex differences in TRD patients by using ALFF and further observed the correlation between clinical symptoms and abnormal brain areas. This study will provide some clinical reference value for the sex differences in neuropathological mechanisms of TRD.

## 2. Methods

### 2.1. Participants

The patients in this study were from Guang’anmen Hospital, China Academy of Chinese Medical Sciences, Xuanwu Hospital of Capital Medical University, and Beijing First Hospital of Integrated Chinese and Western Medicine. We evaluated patients using the Diagnostic and Statistical Manual of Mental Disorders, Fifth Edition (DSM-5) criteria, and included a total of 35 patients with TRD. In addition, patients with TRD were required to meet the following conditions [4]: (1) no significant improvement after two or more full doses and courses of antidepressant treatment; (2) score >17 on the 17-item Hamilton Rating Scale for Depression scale (HAMD-17); (3) right-handedness; (4) age 18–60 years. We also included a total of 39 healthy controls (HCs) relatively matched to the age and sex of the TRD group. The exclusion criteria for the TRD and HC groups were as follows: (1) contraindications to MRI scans; (2) other mental illnesses; (3) presence of traumatic brain injury and other cardiovascular and cerebrovascular diseases; (4) pregnant; (5) history of alcohol addiction.

### 2.2. Scan Acquisition

In this study, all subjects underwent data acquisition using a Magnetom Skyra 3.0-T scanner (Siemens, Erlangen, Germany) at Guang’anmen Hospital, China Academy of Chinese Medical Sciences. Subjects wore noise-cancelling headphones, secured their heads with a hood, closed their eyes to keep their minds clear and avoided active thinking.

The scanning parameters meet the following criteria: For three-dimensional T1-weighted imaging, repetition time (TR)/echo time (TE) = 2500/2.98 ms, flip angle (FA) = 7°, matrix = 64 × 64, slice number = 48, field of view (FOV) = 256 × 256 mm^2^, slice thickness = 1 mm, slices = 192, time = 6 min 3 s. for BOLD, TR/TE = 2000/30 ms, matrix = 64 × 64, slice number = 43, FOV = 240 × 240 mm^2^, FA = 90°, number of obtained volumes = 200, slice thickness/gap = 3.0/1.0 mm, time = 6 min 40 s.

### 2.3. Image Processing

#### 2.3.1. fMRI Data Preprocessing

In this study, the image data were preprocessed using the DPARSF toolkit (DPARSF 5.0, http://www.rfmri.org/DPARSF (accessed on 28 September 2022) [25]. The specific preprocesses are as follows: (1) convert 2D DICOM format images to 3D NII format images; (2) remove the scanned data for the first 10 time points; (3) slice timing; (4) perform head movement correction and remove subjects with head movement >2 mm and rotation >2° after reporting head movement data; (5) use a resolution of 3 mm × 3 mm × 3 mm and spatially normalize all 3D image data according to the standardized space defined by the Montreal Neurological Institute (MNI) defined normalization space, all 3D image data were spatially normalized; (6) the spatially normalized data were smoothed using a 6 × 6 × 6 mm^3^ Gaussian kernel; (7) linear detrending; (8) finally, head movement, cerebrospinal fluid, and cerebral white matter covariates were removed.

#### 2.3.2. ALFF Analysis

The ALFF values were calculated for each study subject using DPARSF software. The voxel power spectrum was obtained by processing the time series of each voxel using Fast Fourier Transform, calculating the square root of each frequency on the power spectrum to obtain the signal oscillation amplitude, calculating the mean value of the power spectrum for each voxel within 0.01–0.08 Hz, and normalizing the mean value of the power spectrum for all voxels in the whole brain to obtain the ALFF value for each brain region.

### 2.4. Statistical Analyses

#### 2.4.1. Clinical Data

Statistical analysis of demographic data was performed using SPSS 23.0 software. Compare age and years of educational among the four groups with one-way analysis of variance (ANOVA). Compare the HAMD-17 scores and duration of disease between the two patient groups with two-sample *t*-test. A statistical threshold setting of *p* < 0.05 was statistically significant.

#### 2.4.2. fMRI Data

Image data were analyzed using DPARSF 5.0 software. To analyze effects due to overall differences in group, sex and their interactions, we entered all voxel-based comparisons of whole-brain ALFF maps into a random-effects 2 (group: TRD, HC) × 2 (sex: male, female) ANOVA model. In addition, we corrected the differences in ALFF among the four groups for Gaussian random fields (GRF) and controlled years of education, framewise displacement (FD) and age as covariates, with a threshold voxel level defined as *p* < 0.005 and a clustering level defined as *p* < 0.05, which was considered statistically significant.

We using DPARSF 5.0 software to extracted ALFF values for differential brain regions for sex-by-group interaction effect. And using SPSS 23.0 performed post hoc two-sample *t*-test analysis between groups, with bonferroni correction for the results and threshold set at *p* < 0.0125 (0.05/4) to be statistically significant.

We used Pearson correlation analysis to verify the relationship between abnormal brain areas and clinical symptoms in different sex of the TRD groups. In addition, and controlling for mean FD value, age, and years of education. The statistical threshold was set at *p* < 0.05 to be statistically significant.

## 3. Results

### 3.1. Characteristics of Research Samples

Because of excessive head movement, two HC patients and one TRD patient were excluded. Therefore, a total of 16 male patients with TRD, 18 female patients with TRD, 18 male HCs and 19 female male HCs met the experimental criteria. Among the four groups in terms of years of education and age were no significant differences. Moreover, there were no significant differences in duration of illness and HAMD-17 scores between the two TRD groups (Table 1).

### 3.2. Main Effects of Group, Sex, and Sex-by-Group Interaction in ALFF among the Four Groups

The main effects of the group difference in ALFF among the four groups was located in the left pallidum (Table 2; Figure 1A).

The main effects of sex difference in ALFF among the four groups was located in the right posterior cerebellar lobe (Table 2; Figure 1B).

The effects of the sex-by-group interaction difference in ALFF among the four groups was located in the left middle frontal gyrus, left precentral gyrus and left precuneus (Table 2; Figure 1C).

### 3.3. Post Hoc Analyses in ALFF among the Four Groups

The male TRD group had lower ALFF in the left middle frontal gyrus and left precentral gyrus compared with the female TRD group. The male TRD group had lower ALFF in the left precuneus compared with the male HC group. The female TRD group had higher ALFF in the left middle frontal gyrus and left precentral gyrus compared with the female HC group. The male HC group had higher ALFF in the left precuneus compared with the female HC group (Figure 2).

### 3.4. Relationships between ALFF and Clinical Symptoms

In the present study, we controlled mean FD values, age, and years of education. The ALFF values of the left middle frontal gyrus and left precuneus in the female TRD group were positively correlated with HAMD-17 scores (*r* = 0.543, *p* = 0.037; *r* = 0.690, *p* = 0.004), while the ALFF values of the left precentral gyrus in the female TRD group were negatively correlated with HAMD-17 scores (*r* = −0.538, *p* = 0.039) (Figure 3).

## 4. Discussion

In this study, we observed the sex differences of the functional brain activity in the TRD groups. From the results, the male TRD group had lower ALFF in the left middle frontal gyrus and left precentral gyrus compared with the female TRD group. In addition, we found a correlation between ALFF values of the left middle frontal gyrus, left precentral gyrus, left precuneus and depression severity in female TRD group. This study provides some implications for the sex differences in neuropathological mechanisms of TRD.

We found that the male TRD group had lower ALFF in the left middle frontal gyrus compared with the female TRD group. The left middle frontal gyrus is part of the dorsolateral prefrontal lobe (DLPFC), and an important component of the CCN [26,27,28]. The DLPFC is responsible for top-down regulation of emotional processing, decision making, working memory, and attention processing, and is closely related to the onset of MDD [29,30,31,32]. DLPFC is an important target for repetitive transcranial magnetic stimulation in the treatment of TRD and is effective in preventing relapse in patients with TRD [33,34]. A study found that the male MDD patients had lower ALFF in the left superior frontal gyrus compared with female MDD patients, which supports the results of this study [17]. Therefore, the results of this study suggest that there are sex differences in the CCN of patients with TRD, which may lead to different barriers to work for patients of different sexes performing the same tasks. Moreover, we found that the ALFF values of the left middle frontal gyrus were positively correlated with HAMD-17 scores in the female TRD group, suggesting that the left middle frontal gyrus may be an important neuroimaging marker in the female TRD group.

In this study, we found that the male TRD group had lower ALFF in the left precentral gyrus compared with the female TRD group. The left precentral gyrus is an important part of the sensorimotor network and a higher center for somatic sensation and movement, involved in executive control and emotion management, and working memory functions [35,36,37]. A study found that the functional abnormalities in the sensorimotor network in patients with MDD [38,39]. Compared to the group without somatic disorder MDD, the somatic disorder MDD group had more severe depressive symptoms and reduced ALFF in the precentral and postcentral gyrus [40]. Previous studies have found structural and functional impairments in the precentral gyrus of patients with TRD [41,42]. The male MDD group had higher ALFF in the left postcentral gyrus compared with the female MDD group, suggesting that MDD patients of different sexes have functional impairment in the sensorimotor network [17]. Therefore, the results of this study suggest that functional abnormalities in the left precentral gyrus may be an important brain regions for distinguishing male TRD from female TRD. Previous studies have found that ALFF values in the left postcentral gyrus of female MDD are positively correlated with the severity of weight loss in patients, suggesting that the left postcentral gyrus may be an important brain region in differentiating the pathogenesis of weight loss in patients with MDD of different sexes [17]. We further found that ALFF values in the left precentral gyrus of the female TRD group were negatively correlated with HAMD-17 scores, suggesting that ALFF values in the left precentral gyrus may be predictive of disease severity in the female TRD group. We speculate that female are more concerned about somatic symptoms in the case of persistent depressed mood.

The precuneus is an important node in the posterior default mode network (DMN) and is involved in contextual memory processing, emotion regulation and visuospatial imagery functions [43,44]. Previous studies have found functional abnormalities in the DMN of patients with TRD [42,45,46]. Patients with TRD treated with Ketamine infusion were able to reverse the functional abnormalities of the precuneus in the DMN [47]. A study found that the male MDD group had higher ALFF in the right precuneus compared to the female MDD group, suggesting abnormalities in the DMN in patients with MDD of different sex [17]. Although this study did not find abnormalities in the DMN in the different sex TRD groups, it showed abnormalities in the DMN in the male TRD group, which further confirms the important role played by the DMN in patients with TRD.

In addition, the main effects of the group difference in ALFF among the four groups was located in the left pallidum. The pallidum plays an important regulatory role in the cortico-striato-thalamo-cortical circuit, involved in perception, attention and emotional expression [48,49]. The pallidum is also an important node closely associated with MDD reward motivation and interest pleasure, receiving information input from the nucleus ambiguus and transmitting this information to downstream targets [50,51]. A study found that the older TRD group had higher ALFF in the left pallidum compared with the HC group [52]. Important target brain areas for deep brain stimulation (DBS) and vagus nerve stimulation (VNS) for TRD include the pallidum. DBS and VNS can improve clinical symptoms of TRD by modulating brain regions that are closely related to the pathogenesis of TRD, including the pallidum [53]. Therefore, the results of this study suggest that the TRD group had functional abnormalities in the left pallidum, which we speculate may be related to the lack of pleasure and loss of interest in TRD.

### Limitations

There are some limitations of this study that need to be noted. First, antidepressants may have an effect on brain function in TRD patients, which is something we cannot ignore. Second, this study focused on a relatively single clinical symptom in TRD patients, focusing on only one scale, the HAMD-17, but not on the anxiety and insomnia scales, and further research is needed in the future. Third, we did not include the nTRD group, which lacks the specificity of TRD neuropathology. Finally, we included a relatively small sample size, which needs to be further expanded in the future to validate our results.

## 5. Conclusions

In summary, we found differences in neuropathological mechanisms between the male TRD group and the female TRD group in some brain regions, particularly evident in the left middle frontal gyrus and left precentral gyrus, with differences associated with abnormalities in the CCN and sensorimotor network. This study may contribute to potential therapeutic targets for the clinical treatment of patients with sex-specific TRD.

## Figures and Tables

**Figure 1 brainsci-12-01604-f001:**
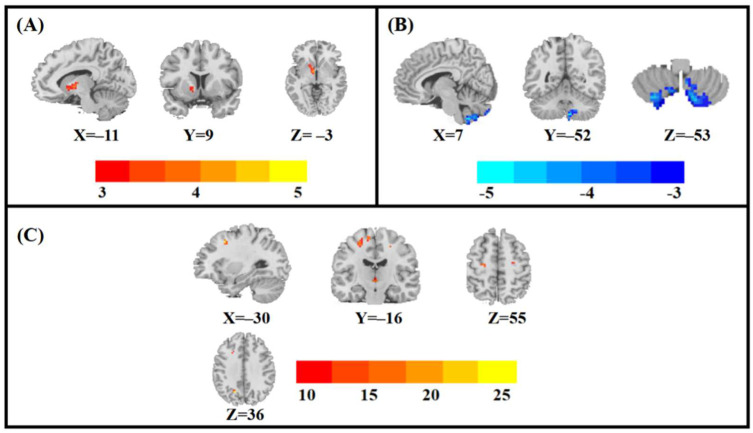
Statistical maps showing significant ALFF differences among the four group. (**A**), Main effects of the group; (**B**), Main effects of sex; (**C**), Sex-by-group interaction effects.

**Figure 2 brainsci-12-01604-f002:**
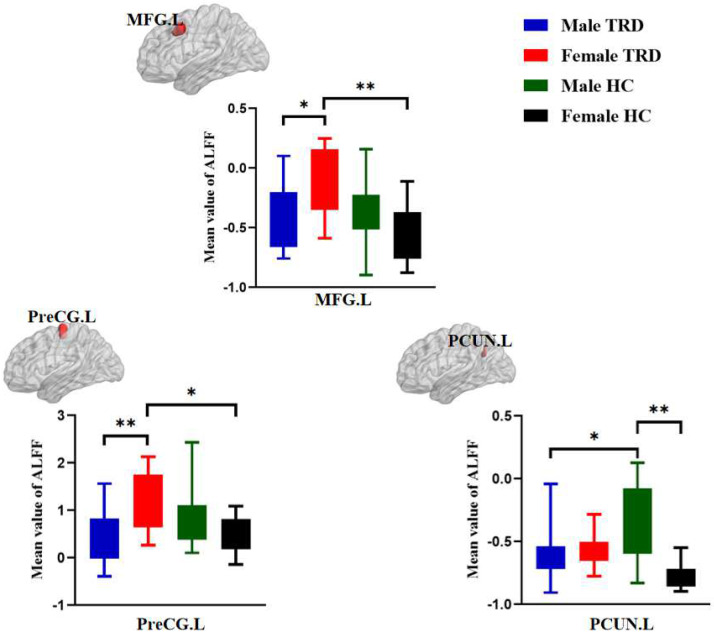
Post hoc analyses in ALFF among the four groups. MFG.L, Left middle frontal gyrus; PreCG.L, left precentral gyrus; PCUN.L, Left precuneus; ** *p* < 0.001, * *p* < 0.0125.

**Figure 3 brainsci-12-01604-f003:**
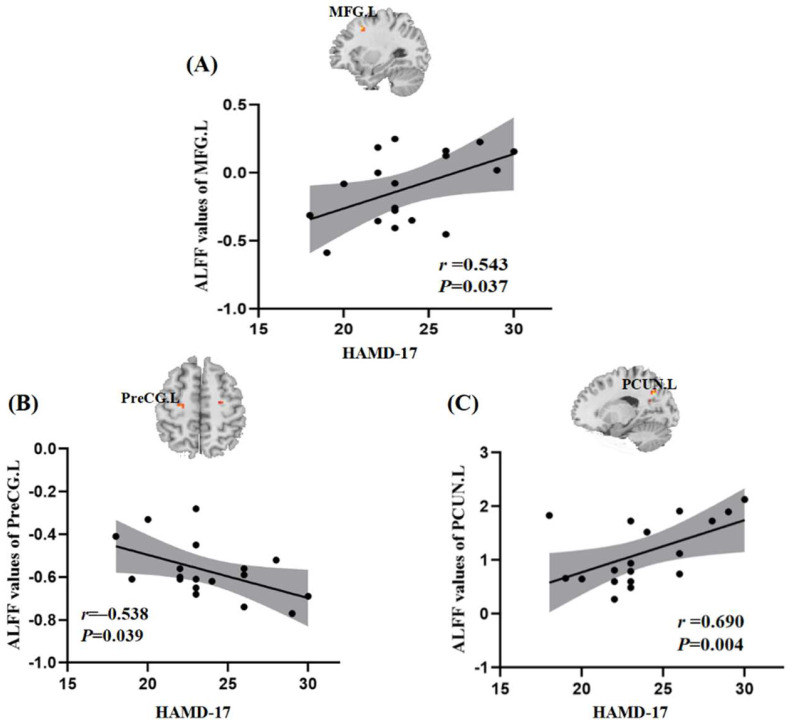
Relationships between ALFF and clinical symptom. Positive correlation between the ALFF values and HAMD-17 scores in the female TRD group: (**A**), MFG.L, Left middle frontal gyrus; (**C**), PCUN.L, Left precuneus. Negative correlation between the ALFF values and HAMD-17 scores: (**B**), PreCG.L, left precentral gyrus. HAMD-17, 17-item Hamilton Rating Scale for Depression.

**Table 1 brainsci-12-01604-t001:** Demographic and clinical characteristics of all of the participants.

Variable	Male TRD (*n* = 16)	Female TRD (*n* = 18)	Male HCs (*n* = 18)	Female HCs (*n* = 19)	*t(F)*/χ^2^	*p*-Value
Age (years)	40.37 ± 8.98	42.16 ± 10.34	41.72 ± 11.06	43.68 ± 10.98	0.300	0.825 ^a^
Education (years)	14.37 ± 2.91	14.05 ± 3.11	14.61 ± 3.16	13.47 ± 4.38	0.369	0.775 ^a^
Duration of illness (months)	46.00 ± 18.40	43.33 ± 14.90	-	-	0.467	0.644 ^b^
HAMD-17 score	22.18 ± 2.73	23.72 ± 3.26	-	-	−1.474	0.150 ^b,^*

TRD, treatment-resistant depression; HCs, healthy controls; ^a^ one-way ANOVA tests. ^b^ two-sample *t*-test. * Significant difference.

**Table 2 brainsci-12-01604-t002:** Brain areas with significant ALFF differences among the four groups.

Clusters	Brain Regions	Peak Coordinates (MNI)			Cluster Size	*T/F*-Values
		X	Y	Z		
*Main effects of the group*						
1	Left pallidum	−11	9	−3	51	3.570 ^a^
*Main effects of sex*						
1	Right posterior cerebellar lobe	7	−52	−53	22	−4.091 ^a^
*Sex-by-group interaction effects*						
1	left middle frontal gyrus	−30	12	45	21	26.370 ^b^
2	left precentral gyrus	−27	−16	55	22	17.058 ^b^
3	left precuneus	−18	−63	36	24	27.699 ^b^

^a^ The P value indicates the T value, ^b^ The P value indicates the F value.

## Data Availability

Data can be made available upon reasonable request.

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
