# Peer review of "Sex Differences of the Functional Brain Activity in Treatment-Resistant Depression: A Resting-State Functional Magnetic Resonance Study"

_brainsci, 2022, doi:10.3390/brainsci12121604_

Round 1
Reviewer 1 Report
The whole manuscript is well-structured and provides sufficient details within every section. There are only some smaller remarks.
- please introduce all abbreviations at first use (e.g. ALFF within the abstract, BOLD etc.)
- the introduction could be enhanced by a short section on BOLD, fMRI and rs-fMRT as technique that is used
- the authors desribe exclusion criteria. later on 3 subjects are excluded due to extensive head movement during data acquisition. This exclusion criteria should be stated in the methods section too.
- the scan acquisition should also encompass paramters of T1-weighted imaging (just for completeness) and information if fMRI data was acquired ascending/descening or interleaved, and if subjects were instructed to have there eye's open or closed as this could impact rs networks too.
Author Response
Manuscript ID: brainsci-2026077
Date: 2022.11.12
Title:Sex differences of the functional brain activity in treatment-resistant depression: A resting-state functional magnetic resonance study
Author name:Ji-fei Sun
Reply to the comments
Dear editor,
We are grateful to the referees for their careful reading of our papers. We have carefully considered the comments and have revised the manuscript accordingly. Please find below our responses to the reviewers’comments. All the revisions have been addressed in the Reply and highlighted in the manuscript with yellow background. We hope the revised manuscript can be considered acceptable.
Reply to the comments of Reviewer 1
Comments:
- please introduce all abbreviations at first use (e.g. ALFF within the abstract, BOLD etc.)
Reply: Many thanks to the reviewers for asking such a rigorous and detailed questionand. Based on the reviewers' comments, we have revised the manuscript.
- the introduction could be enhanced by a short section on BOLD, fMRI and rs-fMRT as technique that is used
Reply: Thank you very much for the experts' helpful suggestions. Based on the reviewers' comments, we have revised the manuscript.
- the authors desribe exclusion criteria. later on 3 subjects are excluded due to extensive head movement during data acquisition. This exclusion criteria should be stated in the methods section too.
Reply: Thank you very much for the experts' helpful suggestions. Based on the reviewers' comments, we have revised the manuscript.In "fMRI data preprocessing", we have added.
- the scan acquisition should also encompass paramters of T1-weighted imaging (just for completeness) and information if fMRI data was acquired ascending/descening or interleaved, and if subjects were instructed to have there eye's open or closed as this could impact rs networks too.
Reply: Many thanks to the reviewers for asking such a rigorous and detailed questionand. We strongly endorse the reviewers' comments, and we have made additions based on the reviewers' comments. And in the manuscript, we further elaborated on T1 and added that subjects need to close their eyes。
Finally, I am very grateful to the reviewers for their authoritative, professional and meticulous help, and we have read through and checked the whole manuscript again. I have gained a lot from the interaction with you. Best wishes to you.

Reviewer 2 Report
Thank you for allowing me to review the paper. I agree with identifying the specific neural mechanisms in order to improve treatment outcomes for TRD. The authors challenged to elucidate the mechanism from "gender differences", by comparing of ALFF values with 2x2 grouping, sex and diagnosis.
Strong points:
The sentences are easy to understand and the design is simple. The validity of the results derived from authors' methodology is also supported by previous studies.
However, concerns remained regarding the purpose of the analysis and its interpretation. In addition, the "Introduction" and "Discussion" sections contain an abundance information that is not directly relevant to this paper and is beyond the scope of the data addressed in this paper. The authors need to tone down the entire document according to their own design.
My specific comments are below.
[Major comments]
#1. at Line 37:
It does not logically derive why it is important to study the "neuropathological" mechanisms of TRD. The immediately prior sentence described the clinical difficulties associated with TRD. The link between these problems and this sentence (lines 37-39) is ambiguous. I believe that many readers would be interested to know why an approach from neuropathological mechanisms is necessary.
#2. at Paragraph 3 in Introduction:
Although previous studies of structural imaging are described in detail, this paper is focused on functional imaging. A broader review should be undertaken on functional analyses other than ALFF (e.g., ReHo, functional connectivity, etc.).
#3. at Line 68:
The results of previous studies are comprehensively described by the network labels. I recommend that the authors include the name of the areas detailed, as this information is important when referring to the results of this paper.
#4. at Paragraph 4 in Introduction:
Some of the studies the authors have already listed in the discussion may also be referred to in the introduction to guide the hypothesis. In addition, please introduce any brain imaging studies about the differences between TRD and MDD (nTRD or TSD). This information is necessary to discuss the subject group with its limitations.
For example,
Lui S, Wu Q, Qiu L, Yang X, Kuang W, et al. (2011) Resting-state functional connectivity in treatment-resistant depression. Am J Psychiatry 168: 642–648.
Guo WB, Liu F, Chen JD, Gao K, Xue ZM, et al. (2012) Abnormal neural activity of brain regions in treatment-resistant and treatment-sensitive major depressive disorder: A resting-state fMRI study. J Psychiatr Res 46: 1366–1373.
#5. at Line 77:
The hypothesis is vague and does not allow the reader to predict the results. Likewise, the process by which this hypothesis is derived is unclear. Please clarify the path to the hypothesis and the content of the hypothesis.
#6. at “Methods” and “Limitations”a:
The control group of healthy subjects makes it unclear whether the differences found are "sex differences in TRD" or "sex differences in depression". The authors approached it at the limitation point, but their worry was not reflected in the interpretation of the results. The discussion should be stated with attention to the comparison with healthy subjects. In particular, given that the authors' results also include characteristics as MDD, please state whether they are as MDD or (may be) TRD by reference to previous studies.
#7. at 2.4.2 "Within-group patterns":
What does this term mean? Two random effects were entered in the analysis, and these were the factors that defined the groups. Therefore, "ALFF differences among the four groups" is correct, but why is this "Within"? One-sample tests per group often reflect this.
Also, what is the rationale for the statistical thresholds in this analysis?
#8. at “Correlation analysis” (method, result and discussion):
I am not certain what the results reveal about ALFF in sex-differentiated regions being correlated with depressive symptoms. Symptom severity is the same between males and females, so why is the association between sex-differentiated brain regions and depressive symptoms being examined? Isn't it necessary to hypothesize that different sexes have different brain regions that exacerbate (not "resist treatment") depressive symptoms?
The result is very simple. Only female patients are shown to be severely aggravated by hyperactivity in this brain region. Please add references that make this interpretable.
Alternatively, the authors can relate it to other indicators instead of gender severity. For example, could you present data that could explain the clinical sex difference as described from line 44? That leads directly to the suggestion in line 48.
#9. at Line 225:
The expanded interpretation labels the findings in a few brain regions as a network. It is not safe to discuss as a whole network with only one specific region within the network.
#10. at Lines 230 & 249:
Are these "reduce" correct? I can see an "increase" in the female patient group when compared to the healthy group.
#11. at Line 241:
These previous studies are not directly related to the results of this paper. Please do not link interpretations based solely on brain region matches, and pay attention to the consistency of measurement and analysis methods.
#12. at Line 262:
This previous study also cannot be related to the present study.
#13. at Line 265:
It is possible to distinguish between male and female patients with TRD without measuring this brain region (using biological labels). The previous studies given by the authors in this paragraph were on MDD. Considering that healthy subjects were the control group, this region's abnormality may depend on diagnosis rather than on "treatment resistance". On that basis, discuss the results of the sex difference.
#14. at Paragraph 4 in Discussion
This paragraph cannot be mentioned from the results in this paper.
Minor comments:
#15. Subjects:
What is the statistical basis for the sample size design?
#16. Term “TRD patients”:
Kindly note that many journals and recent academic conventions encourage the humanization and respect of patients when addressing them by including the word “patient” in the initial term addressing them. For example, “depressed patient” should be ideally phrased as “patient with depression”.
#17. at “3.3 post hoc analyses”:
Please provide t-values and p-values in the text.
#18. at Figure 1:
Include an explanation of the ABC labels in the caption or include them in the figure.
#19. at Figure 2:
Align the ranges on the y-axis.
#20. at Figure 3:
Align the ranges on the y-axis. And, It is better to include the male data as well, so that the results can be understood more accurately.
Author Response
Manuscript ID: brainsci-2026077
Date: 2022.11.12
Title: Sex differences of the functional brain activity in treatment-resistant depression: A resting-state functional magnetic resonance study
Author name:Ji-fei Sun
Reply to the comments
Dear editor,
We are grateful to the referees for their careful reading of our papers. We have carefully considered the comments and have revised the manuscript accordingly. Please find below our responses to the reviewers’comments. All the revisions have been addressed in the Reply and highlighted in the manuscript with green background. We hope the revised manuscript can be considered acceptable.
Reply to the comments of Reviewer 2
Comments:
(1)at Line 37: It does not logically derive why it is important to study the "neuropathological" mechanisms of TRD. The immediately prior sentence described the clinical difficulties associated with TRD. The link between these problems and this sentence (lines 37-39) is ambiguous. I believe that many readers would be interested to know why an approach from neuropathological mechanisms is necessary.
Reply: Many thanks to the reviewers for asking such a rigorous and detailed questionand. We strongly endorse the reviewers' comments. Based on the reviewer's comments, we have deleted the phrase "neuropathological mechanisms" and added the phrase "in addition" in the next paragraph to follow.
(2)at Paragraph 3 in Introduction: Although previous studies of structural imaging are described in detail, this paper is focused on functional imaging. A broader review should be undertaken on functional analyses other than ALFF (e.g., ReHo, functional connectivity, etc.).
Reply: Many thanks to the reviewers for their comprehensive comments. We strongly endorse the reviewers' comments. To our knowledge there are only 2 studies on differences in resting state brain function in MDD patients, [16], [17]. No methods were found regarding ReHo as well as FC. Also we refer to other studies that have addressed to some extent the studies on structural image differences in terms of sex in MDD, and therefore reviewed them, and again we thank the reviewers. Refer to the following related articles:
[16]Mei L, Wang Y, Liu C, et al. Study of Sex Differences in Unmedicated Patients With Major Depressive Disorder by Using Resting State Brain Functional Magnetic Resonance Imaging. Front Neurosci. 2022;16:814410. Published 2022 Mar 31. doi:10.3389/fnins.2022.814410
[17]Yao Z, Yan R, Wei M, Tang H, Qin J, Lu Q. Gender differences in brain activity and the relationship between brain activity and differences in prevalence rates between male and female major depressive disorder patients: a resting-state fMRI study. Clin Neurophysiol. 2014;125(11):2232-2239. doi:10.1016/j.clinph.2014.03.006
(3). at Line 68:The results of previous studies are comprehensively described by the network labels. I recommend that the authors include the name of the areas detailed, as this information is important when referring to the results of this paper.
Reply: Many thanks to the reviewers for asking such a rigorous and detailed questionand. As suggested by the reviewers, we have added details of brain regions in the manuscript.
- . at Paragraph 4 in Introduction:Some of the studies the authors have already listed in the discussion may also be referred to in the introduction to guide the hypothesis. In addition, please introduce any brain imaging studies about the differences between TRD and MDD (nTRD or TSD). This information is necessary to discuss the subject group with its limitations.
Reply: We thank the reviewers for their very comprehensive and critical comments. This study focused on observing the main differences in brain function between TRD patients with respect to sex. Therefore, there is no specific practical significance in introducing the differences between TRD and nTRD in terms of brain function imaging mechanisms in the introduction. In addition, as requested by the reviewers, we have added in limitations to the non-inclusion of the nTRD group. Once again, we thank the reviewers.
- #5. at Line 77: The hypothesis is vague and does not allow the reader to predict the results. Likewise, the process by which this hypothesis is derived is unclear. Please clarify the path to the hypothesis and the content of the hypothesis.
Reply: We thank the reviewers for their very comprehensive and critical comments. Previous studies on TRD in terms of sex are lacking. Therefore, we are unable to formulate a specific hypothesis. Based on the reviewers' comments, we removed the hypothesis to avoid ambiguity. Thus, it is more beneficial to express the ideas of this manuscript clearly.
- . at “Methods” and “Limitations”a:The control group of healthy subjects makes it unclear whether the differences found are "sex differences in TRD" or "sex differences in depression". The authors approached it at the limitation point, but their worry was not reflected in the interpretation of the results. The discussion should be stated with attention to the comparison with healthy subjects. In particular, given that the authors' results also include characteristics as MDD, please state whether they are as MDD or (may be) TRD by reference to previous studies.
Reply: We thank the reviewers for their very comprehensive and critical comments.We strongly endorse the reviewers' recommendations. In the present study, because the nTRD group was not included, we are also speculating only that this could be a possible difference in neuropathological mechanisms that exist in the TRD group across sexes. Also, the ALFF differences in TRD across age groups were published earlier in this study. Again, we thank the reviewers.The references are as follows.
- Sun J, Guo C, Ma Y, et al. A comparative study of amplitude of low-frequence fluctuation of resting-state fMRI between the younger and older treatment-resistant depression in adults.Front Neurosci. 2022;16:949698. Published 2022 Aug 25. doi:10.3389/fnins.2022.949698
- at 2.4.2 "Within-group patterns":What does this term mean? Two random effects were entered inthe analysis, and these were the factors that defined the groups. Therefore, "ALFF differences among the four groups" is correct, but why is this "Within"? One-sample tests per group often reflect this. Also, what is the rationale for the statistical thresholds in this analysis?
Reply: We thank the reviewers for their very comprehensive and critical comments.We will remove the ‘Within-group patterns.’This study used GRF correction with a threshold set at p<0.005, which should be set more rigorously and reasonably at P<0.001. We speculate that this may be related to the sample size. At the same time, we drew on other related literature, which also had threshold settings using P<0.005. This study is the first exploratory study in patients with TRD of different sexes, therefore, in the future we will further expand the sample size and use more rigorous correction methods to improve the scientific value of this study.he references are as follows(GRF:P<0.005):
[1]Yang Y, Cui Q, Pang Y, et al. Frequency-specific alteration of functional connectivity density in bipolar disorder depression. Prog Neuropsychopharmacol Biol Psychiatry. 2021;104:110026. doi:10.1016/j.pnpbp.2020.110026
[2]Chen P, Chen F, Chen G, et al. Inflammation is associated with decreased functional connectivity of insula in unmedicated bipolar disorder. Brain Behav Immun. 2020;89:615-622. doi:10.1016/j.bbi.2020.07.004
[3]Shunkai L, Su T, Zhong S, et al. Abnormal dynamic functional connectivity of hippocampal subregions associated with working memory impairment in melancholic depression [published online ahead of print, 2021 Dec 6]. Psychol Med. 2021;1-13. doi:10.1017/S0033291721004906
[4]Hong L, Zeng Q, Li K, et al. Intrinsic Brain Activity of Inferior Temporal Region Increased in Prodromal Alzheimer's Disease With Hearing Loss. Front Aging Neurosci. 2022;13:772136. Published 2022 Jan 28. doi:10.3389/fnagi.2021.772136
[5]Chang W, Lv Z, Pang X, Nie L, Zheng J. The local neural markers of MRI in patients with temporal lobe epilepsy presenting ictal panic: A resting resting-state postictal fMRI study. Epilepsy Behav. 2022;129:108490. doi:10.1016/j.yebeh.2021.108490
(8). at “Correlation analysis” (method, result and discussion):I am not certain what the results reveal about ALFF in sex-differentiated regions being correlated with depressive symptoms. Symptom severity is the same between males and females, so why is the association between sex-differentiated brain regions and depressive symptoms being examined? Isn't it necessary to hypothesize that different sexes have different brain regions that exacerbate (not "resist treatment") depressive symptoms?The result is very simple. Only female patients are shown to be severely aggravated by hyperactivity in this brain region. Please add references that make this interpretable.Alternatively, the authors can relate it to other indicators instead of gender severity. For example, could you present data that could explain the clinical sex difference as described from line 44? That leads directly to the suggestion in line 48.
Reply: We thank the reviewers for their very comprehensive and critical comments. Indeed, we strongly endorse the reviewers' comments. In this study, there was no significant difference in clinical symptoms between male and female TRD, but there was a direct correlation with clinical symptoms in both female patients, and we performed an analysis of biased correlations. We speculate that this correlation may be is an important neuropathological mechanism for the pathogenesis of the female TRD group. As requested by the reviewers, we added references and further interpreted the manuscript.In addition, we have also made further revisions to line 48.
(9)at Line 225:The expanded interpretation labels the findings in a few brain regions as a network. It is not safe to discuss as a whole network with only one specific region within the network.
Reply: We thank the reviewers for their very comprehensive and critical comments. Based on the reviewers' comments, we have revised the manuscript.
- . at Lines 230 & 249:Are these "reduce" correct? I can see an "increase" in the female patient group when compared to the healthy group.
Reply: Thanks to the reviewers for their very comprehensive suggestions. ines 230 & 249 both compare ALFF for the male TRD group to the female TRD group and do not address the healthy control group. Thanks again to the reviewers.
- #11. at Line 241:These previous studies are not directly related to the results of this paper. Please do not link interpretations based solely on brain region matches, and pay attention to the consistency of measurement and analysis methods.
Reply: Thanks to the reviewers for their very comprehensive suggestions. We thank the reviewers for their comments and we have removed this section based on the reviewers' comments.
- at Line 262:This previous study also cannot be related to the present study.
Reply: Thanks to the reviewers for their very comprehensive suggestions. We thank the reviewers for their comments and we have removed this section based on the reviewers' comments.
- at Line 265:It is possible to distinguish between male and female patients with TRD without measuring this brain region (using biological labels). The previous studies given by the authors in this paragraph were on MDD. Considering that healthy subjects were the control group, this region's abnormality may depend on diagnosis rather than on "treatment resistance". On that basis, discuss the results of the sex difference.
Reply: Thanks to the reviewers for their very comprehensive suggestions. Based on the reviewers' comments, we have revised this manuscript.
- at Paragraph 4 in DiscussionThis paragraph cannot be mentioned from the results in this paper.
Reply: Thanks to the reviewers for their very comprehensive suggestions. Thanks to the reviewers for their very comprehensive and critical suggestions. After a post hoc two-by-two comparison, although there was no significant difference between male TRD and female TRD in the precuneus, it also showed an important role in the pathogenesis of TRD patients with DMN. Thanks again to the reviewers.
- #15. Subjects:What is the statistical basis for the sample size design?
Reply: Thanks to the reviewers for their very comprehensive suggestions. Because TRD patients are indeed difficult to include in the clinic, cases are relatively rare, mostly in patients with mild to moderate MDD. We refer to previous related literature in more than 13 cases. In future studies, we will further expand the sample size to improve the scientific value of this study. Thanks again to the reviewers.The references are as follows:
- Yao Z, Yan R, Wei M, Tang H, Qin J, Lu Q. Gender differences in brain activity and the relationship between brain activity and differences in prevalence rates between male and female major depressive disorder patients: a resting-state fMRI study. Clin Neurophysiol. 2014;125(11):2232-2239. doi:10.1016/j.clinph.2014.03.006
- Term “TRD patients”:Kindly note that many journals and recent academic conventions encourage the humanization and respect of patients when addressing them by including the word “patient” in the initial term addressing them. For example, “depressed patient” should be ideally phrased as “patient with depression”.
Reply:We thank the reviewers for their insightful comments. Based on the reviewers' comments, we have revised the whole manuscript.
(17). at “3.3 post hoc analyses”:Please provide t-values and p-values in the text.
Reply:Thanks to the reviewer's comments, the images in the text already indicate P values, suggesting that t values are not meaningful, and based on previous studies, no problems with t values have been found. Thanks again to the reviewers.
(18)at Figure 1:Include an explanation of the ABC labels in the caption or include them in the figure.
Reply: Thanks to the reviewers for their very comprehensive suggestions. Based on the reviewers' comments, we have revised this manuscript.
- at Figure 2:Align the ranges on the y-axis.
Reply: Thanks to the reviewers for their very comprehensive suggestions. Based on the reviewers' comments, we have revised this manuscript.
- #20. at Figure 3:Align the ranges on the y-axis. And, It is better to include the male data as well, so that the results can be understood more accurately.
Reply: Thanks to the reviewers for their very comprehensive suggestions. Aligning the Y-axis of A,B will affect its aesthetics. At the same time, we found that there was no clinical symptom correlation in the male TRD group
Finally, I am very grateful to the reviewers for their authoritative, professional and meticulous help, and we have read through and checked the whole manuscript again. I have gained a lot from the interaction with you. Best wishes to you.

Round 2
Reviewer 2 Report
I could be satisfied with most of the authors' revised manuscripts and responses. In particular, I believe that lines 246-256 are specific mentions that enhance the potential value of this paper.
However, there is one point where I would like authors to re-examine the need for improvement.
For my comment (10) & authors' response:
It was my understanding that these sentences were referring to results compared between males and females in the TRD group. Nevertheless, I afraid that only this mention will be misused by other readers. It can be read from the figure that the activity of the female TRD group is higher even if not via statistical comparison. Imagine the male TRD group being emphasized as having "reduced" activity in certain brain regions without any correspondence to the figure's data. I would recommend "increased" or "higher" in female TRD group, but at least the authors should explain "lower" rather than "reduce" in male TRD group. If the authors understand the above concerns, I would like to leave it to them to decide whether to give final approval to my proposal.
Author Response
Manuscript ID: brainsci-2026077
Date: 2022.11.16
Title: Sex differences of the functional brain activity in treatment-resistant depression: A resting-state functional magnetic resonance study
Author name:Ji-fei Sun
Reply to the comments
Dear editor,
We are grateful to the referees for their careful reading of our papers. We have carefully considered the comments and have revised the manuscript accordingly. Please find below our responses to the reviewers’comments. All the revisions have been addressed in the Reply and highlighted in the manuscript with green background. We hope the revised manuscript can be considered acceptable.
Reply to the comments of Reviewer 2
Comments:
(1)For my comment (10) & authors' response:
It was my understanding that these sentences were referring to results compared between males and females in the TRD group. Nevertheless, I afraid that only this mention will be misused by other readers. It can be read from the figure that the activity of the female TRD group is higher even if not via statistical comparison. Imagine the male TRD group being emphasized as having "reduced" activity in certain brain regions without any correspondence to the figure's data. I would recommend "increased" or "higher" in female TRD group, but at least the authors should explain "lower" rather than "reduce" in male TRD group. If the authors understand the above concerns, I would like to leave it to them to decide whether to give final approval to my proposal.
Reply: Many thanks to the reviewers for asking such a rigorous and detailed questionand. We strongly endorse the reviewer comments. Based on the reviewers' comments, not only the 10th issue has been revised, but also the full text has been moderately revised and proofread. Changed increase to higher and reduce to lower. Thanks again to the reviewers.
Finally, I am very grateful to the reviewers for their authoritative, professional and meticulous help, and we have read through and checked the whole manuscript again. I have gained a lot from the interaction with you. Best wishes to you.
